# Investigation of the Characteristics of Catalysis Synergy during Co-Combustion for Coal Gasification Fine Slag with Bituminous Coal and Bamboo Residue

**Yixin Zhang** [1,2,†], **Wenke Jia** [3,†], **Rumeng Wang** [3], **Yang Guo** [3], **Fanhui Guo** [3], **Jianjun Wu** [1,3,*] **and Baiqian Dai** [4,5,*]

1   National Engineering Research Center of Coal Preparation and Purification, China University of Mining and Technology, No.1 Daxue Road, Xuzhou 221116, China; yixinzhang@cumt.edu.cn
2   Shandong Xuanyuan Scientific Engineering and Industrial Technology Research Institute Co., Ltd., Longgu, Juye, Heze 274918, China
3   School of Chemical Engineering and Technology, China University of Mining and Technology, No.1 Daxue Road, Xuzhou 221116, China; jwk0812@163.com (W.J.); rumengwang@cumt.edu.cn (R.W.); cumt-guoyang@cumt.edu.cn (Y.G.); cumtgfh@163.com (F.G.)
4   Department of Chemical Engineering, Monash University, Clayton, VIC 3800, Australia
5   Monash Suzhou Research Institute, SIP, Suzhou 215028, China
*   Correspondence: jjuw@163.com (J.W.); Bai-Qian.Dai@monash.edu (B.D.); Tel.: +86-139-5135-0506 (J.W.); +61-45-077-1204 (B.D.)
†   Yixin Zhang and Wenke Jia contributed equally to this work.

**Abstract:** As a kind of solid waste from coal chemical production, the disposal of coal gasification fine slag poses a certain threat to the environment and the human body. It is essential for gasification slag (GS) to realize rational utilization. GS contains fewer combustible materials, and the high heating value is only 9.31 MJ/Kg, which is difficult to burn in combustion devices solely. The co-combustion behavior of the tri-fuel blends, including bituminous coal (BC), gasification slag (GS), and bamboo residue (BR), was observed by a thermogravimetric analyzer. The TGA results showed that the combustibility increased owing to the addition of BC and BR, and the ignition and burnout temperatures were lower than those of GS alone. The combustion characteristics of the blended samples became worse with the increase in the proportion of GS. The co-combustion process was divided into two main steps with obvious interactions (synergistic and antagonistic). The synergistic effect was mainly attributed to the catalysis of the ash-forming metals reserved with the three raw fuels and the diffusion of oxygen in the rich pore channels of GS. The combustion reaction of blending samples was dominated by O1 and D3 models. The activation energy of the blending combustion decreased compared to the individual combustion of GS. The analysis of the results in this paper can provide some theoretical guidance for the resource utilization of fine slag.

**Keywords:** coal gasification fine slag; co-combustion; catalytic interaction; reaction kinetic

## 1. Introduction

Coal gasification is a core technology for the clean and effective utilization of coal. Coal gasification fine slag, a type of solid waste, is produced in the gasification process. The amount of fine slag adds up to hundreds of millions of tons per year [1,2]. Currently, the disposal of gasification fine slag is mainly by stacking and landfilling [3], which not only causes an increase in transportation expense but also soil and water pollution caused by the leachate from fine slag [4]. Consequently, it is highly imperative and significant for fine slag to achieve quantity reduction, harmless treatment, and resource utilization. The scale utilization of coal gasification fine slag on construction materials and ecological treatment is hindered owing to the high carbon content of fine slag. In terms of resource utilization, researchers have paid extensive attention to the development and utilization of

carbon materials, and the preparation of ceramic materials and aluminum/silicon-based products [5]. However, large-scale consumption and utilization, and the high-value-added conversion of large quantities still lack engineering experience; they are all still in the stage of laboratory research or expanding tests. Shang et al. pointed out that the utilization as blended combustion with circulating fluid-bed raw material was the most popular approach for slag utilization [6]. Dai et al. investigated the co-combustion characteristics of gasification fine slag with bituminous coal, and they concluded that the co-combustion of two fuels showed a nonnegligible synergistic effect. With the increasing blending proportion of bituminous coal, the overall combustion characteristics rose [7]. Wang et al. concluded that Shenhua raw coal blended with a different percent of fine slag could improve the overall combustibility of mixed coal to some extent [8]. Guo et al. studied the physicochemical properties of the residue carbon from froth flotation and indicated that the comprehensive combustion performance was improved with the addition of sawdust char, and the interaction (antagonistic and synergistic) was discovered in the binary blending combustion [9].

Bamboo residues have been discarded during felling and processing. There are many resourceful uses of bamboo residue, for example, wood-based panels, bamboo-plastic composites, bamboo charcoal, bamboo vinegar, and bamboo extracts [10]. Bamboo residue has the advantages of high volatile matter, low nitrogen and sulfur content, and high heat value. If bamboo residues can be fully developed and utilized as green biomass energy, it can largely alleviate the problems of energy shortage in China. Moreover, compared with traditional energy sources, biomass energy is low cost, green, and environmentally friendly, and it can meet the requirements of green development. Direct co-combustion is the simplest, cheapest, and most common option for biomass energy utilization [11]. Christopher et al. studied the potential of co-combustion rice husk and bamboo with coal, and it was noted that the much higher volatile matter content in the biomass fuels played a key role in improving the combustion performance in the system [12]. Hu et al. investigated the combustion behavior of three bamboo residues, and the relatively high HHV and lower N/S contents of the three bamboo residues pointed to their great potential as a clean and renewable feedstock for energy generation [13].

Gasification fine slag cannot be normally burned in the boiler alone, as a result of high water content, low volatile matter, and poor combustion performance [14]. Bamboo residue exhibits excellent combustion features, and co-combustion with fine slag can compensate for the combustion characteristics of fine slag. Wu et al. confirmed that the inorganic constituents (Ca-Fe and Fe oxides) in gasification fine slags could exert a prominent catalytic action on the carbon gasification [15]. Alkali and alkaline earth metal (AAEM) species are the main components of the ash of biomass, and the AAEMs could serve as the catalyst and would affect the combustion of biomass and coal during the co-firing process [16,17]. Zhang et al. adopted a single-particle combustion method, studied the effects of K, Na, Ca, and Mg on the combustion characteristics of pine sawdust and bituminous coal, and found that K played the strongest promotion effect in co-combustion [18]. Economically and environmentally, co-combustion coal with other sources of energy is regarded as a prospective and attractive choice [19,20]. The previous research has predominantly concentrated on binary blending [21–26]. Few studies have focused on the combustion features of tri-blending. Wang et al. investigated the combustion characteristics and kinetics of bituminous coal with pyrolyzed semi-coke and gasified semi-coke [27]. Liao et al. conducted the tri-combustion process for coal, biomass, and polyethylene. The kinetic analysis showed that co-combustion mechanisms followed the diffusion model. The synergy between the tri-fuel was deemed to be governed by the catalytic influence of biomass and the initial exothermic release of energy and free radicals from three fuels [28]. Similarly, the research of tri-blending has also been reported [29–31].

In this paper, we proposed the blending of coal gasification fine slag with bamboo residue and bituminous coal to improve the combustion characteristics of fine slag. This is also beneficial for the direct combustion of bamboo residue as it has a low heating value.

This is beneficial for the sole energy reliability for coal combustion, as BR can be used as an additive. In addition, parts of ash-forming metals reserved in the three raw materials can exert a catalytic effect in the co-combustion process and promote the combustion performance, aiming to change the blending ratio of GS in the process of cyclic sintering. This paper demonstrates the co-combustion characteristics of tri-blends by TG-DTG. It can help to further understand the reaction mechanism and kinetic behavior of co-combustion using the Coats–Redfern model.

## 2. Results and Discussion

### 2.1. Physical and Chemical Characteristics of Samples

#### 2.1.1. Basic Properties

The basic characteristics of the samples are presented in Table 1. The GS has a high carbon content, indicating that GS contains abundant unburned carbon and has the potential to burn as fuel. Nevertheless, GS cannot maintain normal combustion, due to its high ash, low volatile matter, and low HHV. It has been suggested that the high-efficiency utilization of carbon resources be realized by cyclic mixing combustion [32]. $V_d/(V_d + FC_d)$ is defined as the volatile fuel ratio. BR has a higher value than BC and GS do in the volatile matter and volatile fuel ratio, showing that BR behaves well in combustibility. BR can reduce the release of nitrogen oxides and sulfur dioxide due to the low nitrogen and sulfur content, and it can be considered as a clean fuel. The high H/C and O/C are favorable for ignition and combustion intensification. The whole combustion performance may be improved by blending the high-quality fuel (BC and BR) into GS.

**Table 1.** Basic properties of samples.

| Samples | Proximate Analysis (wt%) | | | | Ultimate Analysis (wt%) | | | | | HHV (ad, MJ/kg) |
|---------|-------|-------|-------|--------|-----------|-----------|-------------|-----------|-----------|------|
|  | $M_{ad}$ | $A_d$ | $V_d$ | $FC_d$ | $C_{daf}$ | $H_{daf}$ | $O_{daf}$ * | $N_{daf}$ | $S_{t,d}$ |  |
| BC | 1.62 | 9.33 | 21.89 | 68.77 | 89.23 | 4.78 | 3.85 | 1.48 | 0.61 | 32.83 |
| GS | 3.48 | 71.95 | 3.81 | 24.23 | 93.54 | 0.81 | 3.07 | 0.33 | 0.63 | 9.31 |
| BR | 6.57 | 3.35 | 80.67 | 15.98 | 54.84 | 5.74 | 38.81 | 0.54 | 0.07 | 21.47 |

Notes: d: dry basis; ad: air-dry basis; daf: dry ash-free basis; *: by difference.

#### 2.1.2. Ash Chemical Compositions

Ash chemical compositions of GS, BC, and BR are listed in Table 2. The alkali metals, alkali earth metals, and transition metal oxides of samples in BR account for 75.93%. According to the previous study [15,33,34], the AAEMs and transition metal oxides in biomass can contribute a catalytic effect to the blending combustion process. In addition, the Fe and Ca oxides of GS and the alkali metals of coal can also show catalysis and promote the combustion process. The melting characteristics of the inorganic components in GS are correlated with the separation and utilization of the residue carbon. The high ash melting temperature is not conducive to full decomposition and combustion of the carbon fraction, so GS cannot be used for blending sintering. The total amount of metals with catalytic effect in BC and GS in each case was calculated based on 100 g of the blend, as shown in Table 3.

**Table 2.** Ash chemical composition of samples (wt%).

| Samples | $K_2O$ | $Na_2O$ | $SiO_2$ | $Al_2O_3$ | $Fe_2O_3$ | CaO | MgO | $SO_3$ | $TiO_2$ | $MnO_2$ | $P_2O_5$ |
|---------|--------|---------|---------|-----------|-----------|------|------|--------|---------|---------|----------|
| BC | 0.75 | 0.28 | 52.52 | 37.03 | 3.45 | 1.56 | 0.66 | 1.66 | 1.66 | 0.01 | 0.16 |
| GS | 1.26 | 1.93 | 46.90 | 18.81 | 11.46 | 11.31 | 3.88 | 2.44 | 0.80 | 0.21 | 0.15 |
| BR | 58.97 | 0.46 | 24.17 | 2.26 | 3.11 | 8.47 | 4.11 | 4.42 | 0.17 | 0.64 | 4.07 |

**Table 3.** The amounts of metals with catalytic effect in each case.

| Samples | 5% | 10% | 15% | 25% | 30% | 40% | 50% | 60% | 75% |
|---|---|---|---|---|---|---|---|---|---|
| BC + GS | 1.71 | 2.79 | 3.93 | 6.21 | 7.24 | 9.42 | 11.64 | 13.77 | 17.18 |
| BR | 30.37 | 30.37 | 26.58 | 18.98 | 22.78 | 22.78 | 18.98 | 22.78 | 11.39 |

### 2.1.3. Micromorphology Structure

Figure 1 shows the surface morphologies of BC, GS, and BR. It can be seen in Figure 1(A1–A3) that the BC surface is covered with scaly particles. GS consists of mainly some spherical ash particles melted at high temperatures and loose and porous residue carbon, suggesting that it can provide favorable conditions for oxygen diffusion [35]. As shown in Figure 1(B3), there are four main distribution relationships between ash particles and residue carbon: completely separated from the residue carbon, on the surface of the residual carbon, filled in the pores of the residue carbon matrix, and melted together [36]. Figure 1(C2,C3) demonstrate that numerous bamboo fibers exist in bamboo residue. As the bamboo shavings are composed of different parts of bamboo, the particles identified in the bamboo residue sample differ in morphology. They are intertwined with each other in the form of filaments or rods. Such microscopic morphology poses difficulties for fragmentation.

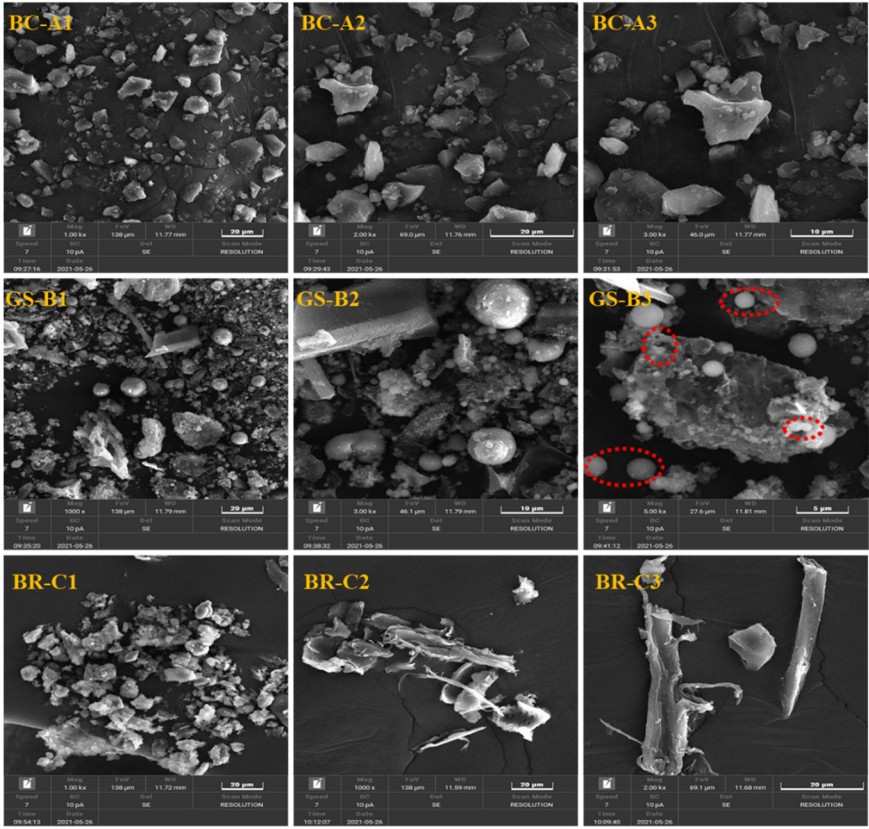

**Figure 1.** SEM images of bituminous coal (**A1–A3**), coal gasification fine slag (**B1–B3**), and bamboo residue (**C1–C3**).

### 2.2. Combustion Characteristic Analysis

### 2.2.1. Combustion Behavior of Individual Fuel

Combustion characteristic parameters are used to evaluate combustion performance. The ignition temperature ($T_i$) reflects the ease of combustion of the sample, and the burnout temperature ($T_b$) indicates the temperature and burnout rate of the sample during the combustion reaction interval, both of which are determined by the tangent method [26].

The comprehensive combustion index (*S*) indicates the overall combustion performance. *S* is calculated by the following equations [8]:

$$S = \frac{\left(\frac{dW}{dt}\right)_{max} \times \left(\frac{dW}{dt}\right)_{mean}}{T_i^2 \times T_b} \tag{1}$$

where $(dW/dt)_{max}$ is the maximum mass loss rate, $(dW/dt)_{mean}$ is the average mass loss rate, $(dW/dt)_{mean} = [(dW/dt)_i + (dW/dt)_b]/2$, and $T_{max}$ denotes the peak temperature.

It could be found from Figure 2A that a mass increase step in the initial stage of BC combustion is relative to the chemical adsorption of oxygen. The DTG curve of BC shows a single wide peak, which is caused by the release of volatiles and the char combustion. The mass loss rate of GS is only about 75%, suggesting that the individual combustion is characterized by high ignition and burnout temperatures. The combustible fraction of GS is only unburned carbon and a small amount of volatiles, and then the DTG curve presents a single peak at around 500–650 °C.

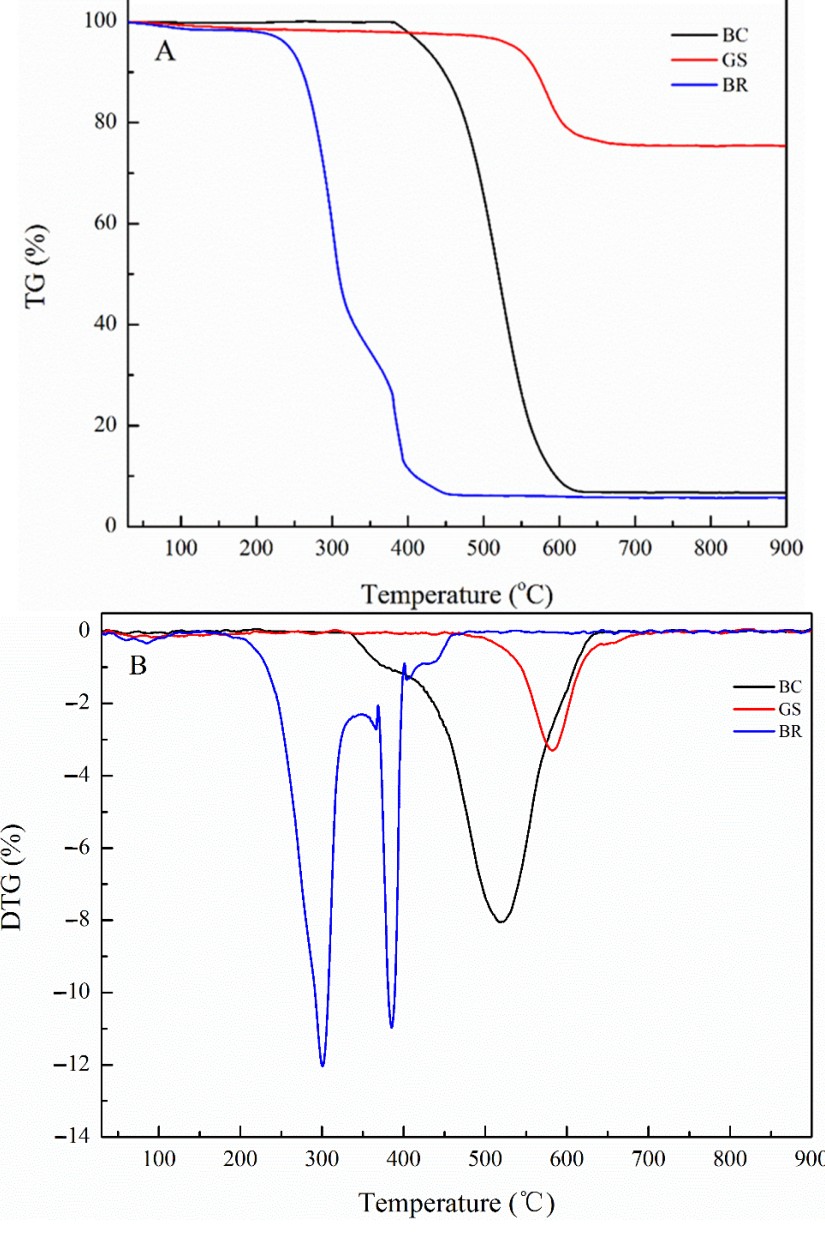

**Figure 2.** The TG (**A**) and DTG (**B**) profiles of raw fuels (BC, GS, and BR).

The combustion of BR is mainly classified as the water evaporation stage, the devolatilization stage, and the char combustion stage [24]. As can be seen from Figure 2B, two sharp peaks of weight loss can be observed at around 200–400 °C. The peak standing for the decomposition of hemicellulose in BR is at around 200–350 °C. The subsequent thermal decomposition of cellulose and parts of lignin occurs at about 350–400 °C. In stage 380–460 °C, the DTG peak is assigned to the combustion reaction of lignin and biochar. The DTG result is consistent with literature reports [11]. It is obviously found from Table 4 that BR has lower ignition and burnout temperatures, and a higher comprehensive combustion index in comparison to BC and GS.

**Table 4.** Combustion characteristic parameters of raw fuels.

| Samples | $T_i$ (°C) | $T_b$ (°C) | $T_{max}$ (°C) | | $(dw/dt)_{max}$ (%/min) | $(dw/dt)_{mean}$ (%/min) | $S$ ($10^{-7}$) |
| | | | Stage 1 | Stage 2 | | | |
|---|---|---|---|---|---|---|---|
| BC | 458.64 | 571.52 | 520.16 | / | 8.07 | 3.25 | 2.18 |
| GS | 540.53 | 614.23 | 580.84 | / | 2.99 | 1.12 | 0.19 |
| BR | 267.59 | 401.85 | 301.29 | 385.33 | 12.8 | 2.93 | 13.03 |

2.2.2. Tri-Fuel Co-Combustion Characteristic

The TG-DTG curves for the tri-fuel blends with different proportions are illustrated in Figure 3A,B. It could be observed that the DTG curves of each blending sample display a two-stage combustion process. From the result shown in Figure 3B, a slight fluctuation occurs at around 46–132 °C, which is attributed to the dewatering drying process of the samples. The volatile matter in BC and BR is decomposed in stage 1. Stage 2 takes place at 349–593 °C and is caused by the combustion of char and combustible material of GS. With the rising blending proportion of GS, the mass loss rate gradually follows an increasing trend, the weight loss peak of the DTG curve appears to shift to higher temperatures from the overall view, and the peak shape becomes wider and shorter. The datas in Table 5 show the tri-blending combustion parameters. An increasing percentage of GS diminishes the comprehensive combustion index S, indicating that with the increase in the addition of GS, the combustibility and combustion features of blending samples become poor. In terms of the combustion data of GS and blends, the co-combustion could effectively improve the individual combustion characteristic.

**Table 5.** Combustion characteristic parameters of various blended samples.

| Samples | $T_i$ (°C) | $T_b$ (°C) | $T_{max}$ (°C) | | $(dw/dt)_{max}$ (%/min) | $(dw/dt)_{mean}$ (%/min) | $S$ ($10^{-7}$) |
| | | | Stage 1 | Stage 2 | | | |
|---|---|---|---|---|---|---|---|
| 5% GS | 264.17 | 534.25 | 290.93 | 487.98 | 4.81 | 2.17 | 2.84 |
| 10% GS | 260.08 | 542.61 | 299.52 | 503.74 | 4.38 | 1.83 | 2.18 |
| 15% GS | 264.99 | 544.59 | 296.61 | 495.54 | 4.51 | 1.80 | 2.12 |
| 25% GS | 269.54 | 553.20 | 301.81 | 499.58 | 4.67 | 1.40 | 1.63 |
| 30% GS | 268.68 | 544.74 | 303.65 | 493.78 | 3.78 | 1.59 | 1.50 |
| 40% GS | 266.40 | 542.04 | 308.88 | 498.33 | 3.07 | 1.27 | 1.02 |
| 50% GS | 286.61 | 557.68 | 316.39 | 505.07 | 2.94 | 1.21 | 0.78 |
| 60% GS | 288.27 | 545.70 | 318.10 | 489.40 | 1.77 | 1.39 | 0.54 |
| 75% GS | 298.37 | 565.63 | 319.89 | 508.61 | 2.10 | 0.92 | 0.38 |

*2.3. Interaction and Synergistic Catalysis of Tri-Fuel Combustion*

The study of interactions between tri-fuels is an important step toward a deeper understanding of the co-firing process. The interaction of tri-fuel was assessed by the difference between the theoretical and experimental conversion rates. The relevant formulas are as follows [9,28,37]:

$$X = \frac{m_i - m_t}{m_i - m_f} \tag{2}$$

$$X_T = X_{BR}Y_{BR} + X_{GS}Y_{GS} + X_{BC}Y_{BC} \qquad (3)$$

$$D = X_E - X_T \qquad (4)$$

In Equations (2)–(4), $X$ is the sample combustion conversion; $m_i$ is the initial mass of the sample; $m_t$ is the mass of the sample at time $t$; $m_f$ is the final mass of the sample; $X_T$ and $X_E$ represent the theoretical and experimental conversion rates, respectively; $D$ denotes the deviation. The $D$ value is greater than zero, which means that there is synergy between the tri-fuel combustion. If $D$ is close to zero or equal to zero, it proves that there is no interaction in the tri-combustion process, the co-combustion is independent. Then, $D$ is less than zero, and the tri-fuel co-combustion presents antagonism or weak interaction.

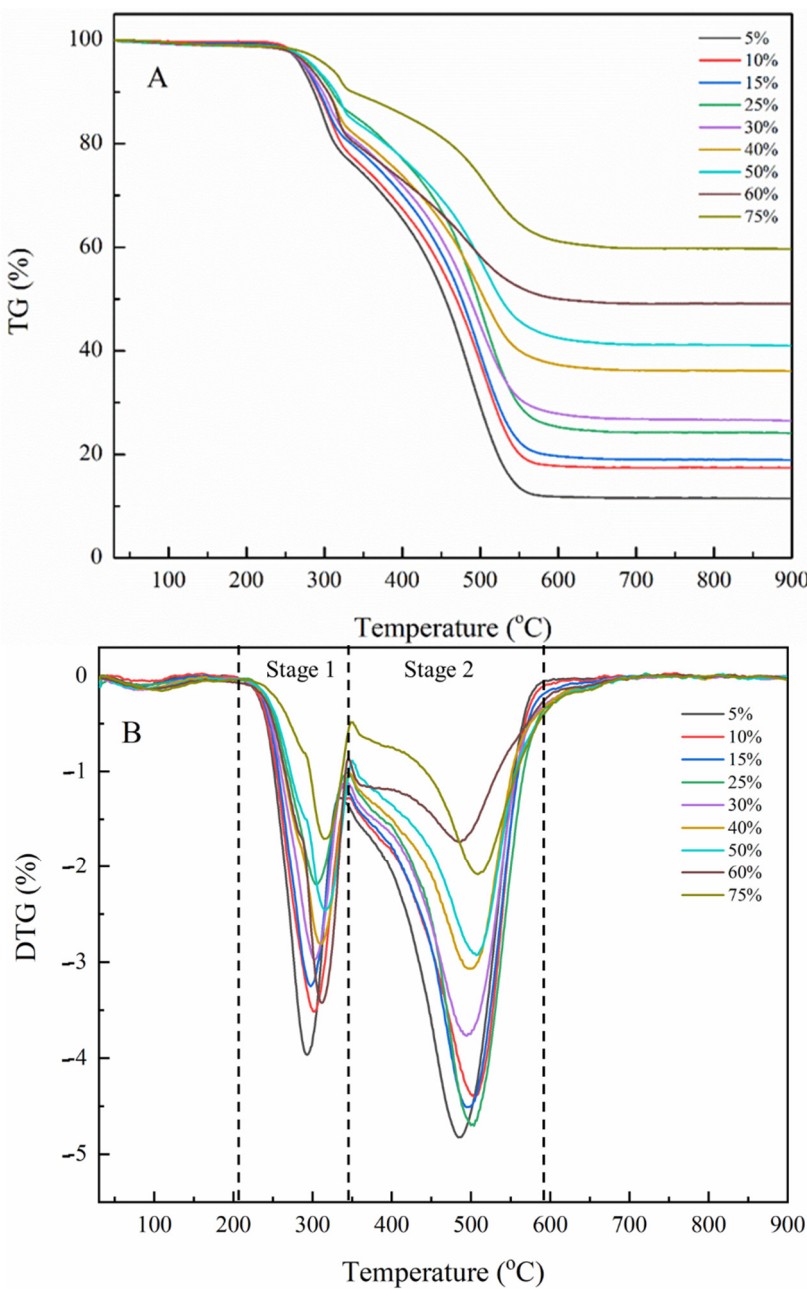

**Figure 3.** TG (**A**) and DTG (**B**) curves of various proportions of samples.

The difference in the theoretical and experimental conversions and the variations in interactions within tri-fuel during combustion can be seen clearly in Figure 4A,B. As depicted in Figure 4A, the difference in $X_T$ and $X_E$ in tri-fuel combustion shows a greater

trend with the growing ratio of GS blended. There are both synergistic and antagonistic effects in the co-combustion process from Figure 4B. When the percentage of GS is 5%, 10%, 15%, and 25%, the co-combustion process is first antagonistic and then synergistic. This may be explained by the fact that due to the high addition of BR in the first four cases, the self-combustion of BR and reaction of the alkali metal and alkaline earth elements in BR with the minerals in BC can result in slagging or agglomerating, hindering the volatile release of BC. The synergistic effect appears after 450 °C. The synergistic effect at this stage is mainly due to the catalytic effect of alkali metals in BC and Fe-Ca oxides in GS. It can be visualized from Table 3 that the catalytic metal content of BC and GS gradually increases with the GS ratio. As shown in Figure 4B, the synergistic effect in each case shows an increasing trend with the increase in the catalytic metal content in the blend, indicating that synergistic catalysis occupies an important role in the combustion reaction. Furthermore, the rising addition of GS has a clear enhancement effect for the synergism. In accordance with the previous study by our group [36], GS is affluent in pores and has a large specific surface area, which facilitates the diffusion and adsorption of oxygen, thus causing a synergistic effect in the reaction. When the addition ratio of GS exceeds 25%, the synergistic effect plays a crucial part in the blending combustion. The synergy between 300 and 400 °C is mainly determined by the AAEMs in BR and the pores of GS. BR is rich in alkali metal and alkaline earth elements. AAEMs could serve as carriers of oxygen, and further promote the migration of oxygen and the combustion of BC and GS. The abundant pore channel provides the paths of oxygen transition. Thus far, the interactions concerning the co-combustion process have no consistent agreement, and the details could be associated with the material properties and reaction conditions.

*2.4. Kinetics Analysis*

Kinetics analysis was adopted to study the physical changes of substances and the rate mechanism of chemical reactions. The model-based method, the Coats–Redfern method, was applied in this paper to acquire kinetic parameters. The model formula is as follows [9,28,38]:

$$\ln\left[\frac{g(x)}{T^2}\right] = \ln\left(\frac{AR}{\beta E}\right) - \frac{E}{RT} \tag{5}$$

where $g(x)$ is the reaction mechanism model function in Table 6, $A$ is the pre-exponential factor, $E$ is reaction activation energy (kJ/mol), $R$ is the gas constant with the value of 8.314 J/(K·mol), and $\beta$ is the heating rate (°C/min). Therefore, a straight line can be obtained when plotting $\ln\left[\frac{g(x)}{T^2}\right]$ versus $1/T$, the straight line is linearly fitted, and the mechanism function of the line with the highest correlation coefficient ($R^2$) is considered to be the reaction mechanism at that stage. The activation energy of the combustion reaction can be calculated by the slope of the line.

**Table 6.** The reaction mechanism model function of $g(x)$.

| Mechanism and Model | $g(x)$ |
| --- | --- |
| Reaction order | |
| O1 | $-\ln(1-x)$ |
| O2 | $(1-x)^{-1}$ |
| O3 | $(1-x)^{-2}$ |
| Phase boundary controlled reaction | |
| R2 | $1-(1-x)^{1/2}$ |
| R3 | $1-(1-x)^{1/3}$ |
| Diffusion models | |
| D1 | $x^2$ |
| D2 | $(1-x)\ln(1-x)+x$ |
| D3 | $\left[1-(1-x)^{1/3}\right]^2$ |
| D4 | $1-2x/3-(1-x)^{2/3}$ |

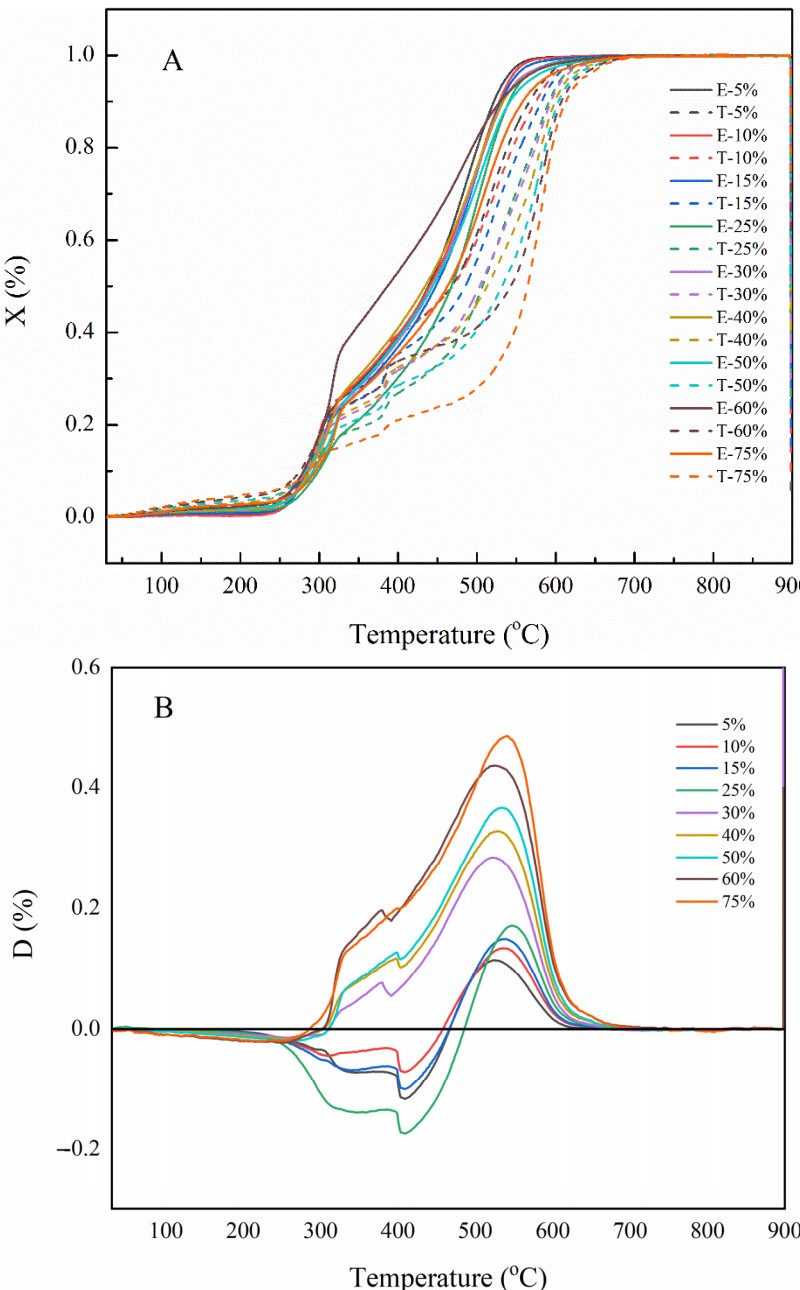

**Figure 4.** (**A**) Comparison of the theoretical and experimental conversion curves; (**B**) Variation profiles of interaction index D.

Table 7 shows the kinetic analysis of raw fuels based on the Coats–Redfern method. The order of the chemical reaction (O1 and O3) is the most optimal mechanism for the combustion reaction of BC and BR [39]. The reaction of GS is controlled by the diffusion model (D3), where D3 is Jander's equation for the diffusion-controlled reaction in a sphere [40]. In the diffusion reaction, most chemical reactions are accomplished by the transportation of gases in the solid phase, that is, the oxygen movement in the pore of residue carbon and GS [9]. The activation energy of the first step of BR is relatively low as a result of the catalytic effect of abundant alkali metals. The activation energy of GS has a higher value, showing that there are many obstacles in the combustion reaction. This result reflects the poor combustibility of GS from the side, which is compatible with the results of the above analysis.

**Table 7.** Combustion kinetic parameters of raw fuels.

| Samples | Stage | E (kJ/mol) | g(x) | $R^2$ |
|---------|-------|------------|------|-------|
| BC | S1 | 43.68 | O1 | 0.9941 |
| GC | S1 | 130.87 | D3 | 0.9718 |
| BR | S1 | 18.14 | O1 | 0.9807 |
| | S2 | 149.37 | O3 | 0.9595 |

As discussed in Section 3.2.2, the co-combustion process is divided into two main stages. Each stage should be analyzed and fitted separately to select the most suitable mechanism function. The kinetic parameters calculated by the subsection calculation method according to the Coats–Redfern model are detailed in Table 8. The combustion reaction of the first stage follows the O1 model ($R^2 > 0.99$), which is controlled by first-order chemical reaction. The reaction of the second stage conforms to the D3 model ($R^2 > 0.98$). This means that the reaction is considered diffusion-controlled. During the co-combustion process, the first stage is mainly assigned to the combustion of the volatile fraction in BR and volatile matter in BC, and then, the combustion of char generated from the previous reaction and residue carbon in GS takes place in the second stage. The activation energy of stage 1 tends to decrease, which probably explains that the interactions between the first four experimental cases are dominated by antagonism in stage 1, the combustion reaction rate is restrained, and the combustion process is also inhibited; consequently, the activation energy of the combustion process is increased. It is notable that the activation energy of stage 2 is higher than that of stage 1. The first stage of the reaction is mainly the combustion of volatile matter from BR and BC. Alkali/alkaline earth metal elements, as active carriers of oxidative adsorption, promote the diffusion of oxygen, increase the oxygen concentration on the surface of carbon atoms, and then promote the combustion reaction [41]. The combustion of BR and BC produces char and molten material, as well as volatile gases, which may block the pores of residue carbon and impede the diffusion of oxygen; the decomposition of combustible materials and carbon components is limited. In addition, the surface of residue carbon is covered by some inorganic minerals melted at high temperatures in GS, restricting the reaction of carbon and the contact of oxygen. Comparatively, the second stage has more difficulty in the combustion reaction, and the activation energy is higher.

**Table 8.** The optimal combustion kinetic parameters of various blending samples.

| Samples | Stage 1 | | | Stage 2 | | |
|---------|---------|---|---|---------|---|---|
| | g(x) | E (kJ/mol) | $R^2$ | g(x) | E (kJ/mol) | $R^2$ |
| 5% GS | O1 | 22.88 | 0.9946 | D3 | 37.45 | 0.9966 |
| 10% GS | OI | 22.81 | 0.9935 | D3 | 44.46 | 0.9906 |
| 15% GS | O1 | 21.37 | 0.9960 | D3 | 37.23 | 0.9840 |
| 25% GS | O1 | 17.37 | 0.9990 | D3 | 41.82 | 0.9865 |
| 30% GS | O1 | 16.02 | 0.9826 | D3 | 34.42 | 0.9909 |
| 40% GS | O1 | 17.96 | 0.9980 | D3 | 30.87 | 0.9838 |
| 50% GS | O1 | 16.15 | 0.9991 | D3 | 31.32 | 0.9843 |
| 60% GS | O1 | 17.88 | 0.9965 | D3 | 20.44 | 0.9946 |
| 75% GS | O1 | 13.56 | 0.9967 | D3 | 35.73 | 0.9909 |

For further illustrating the catalytic role of iron in the co-combustion process, SEM-EDS images of 5% GS, 30% GS, and 75% GS cases are provided in Figure 5. It can be seen that the activation energy decreases gradually for 5%, 30%, and 75% cases from Table 8. The EDS results of the blending samples show a sequential increase in Fe content. The catalytic effect on the combustion process is strengthened with the increase in Fe content [42]. The electron-donating effect of Fe is transferred to the carbon ring or chain by oxygen, making it unstable and cracking, accelerating the process of carbon gasification and improving the

reaction rate of combustion. Therefore, the activation energy shows a decreasing trend due to the catalytic effect of Fe [43].

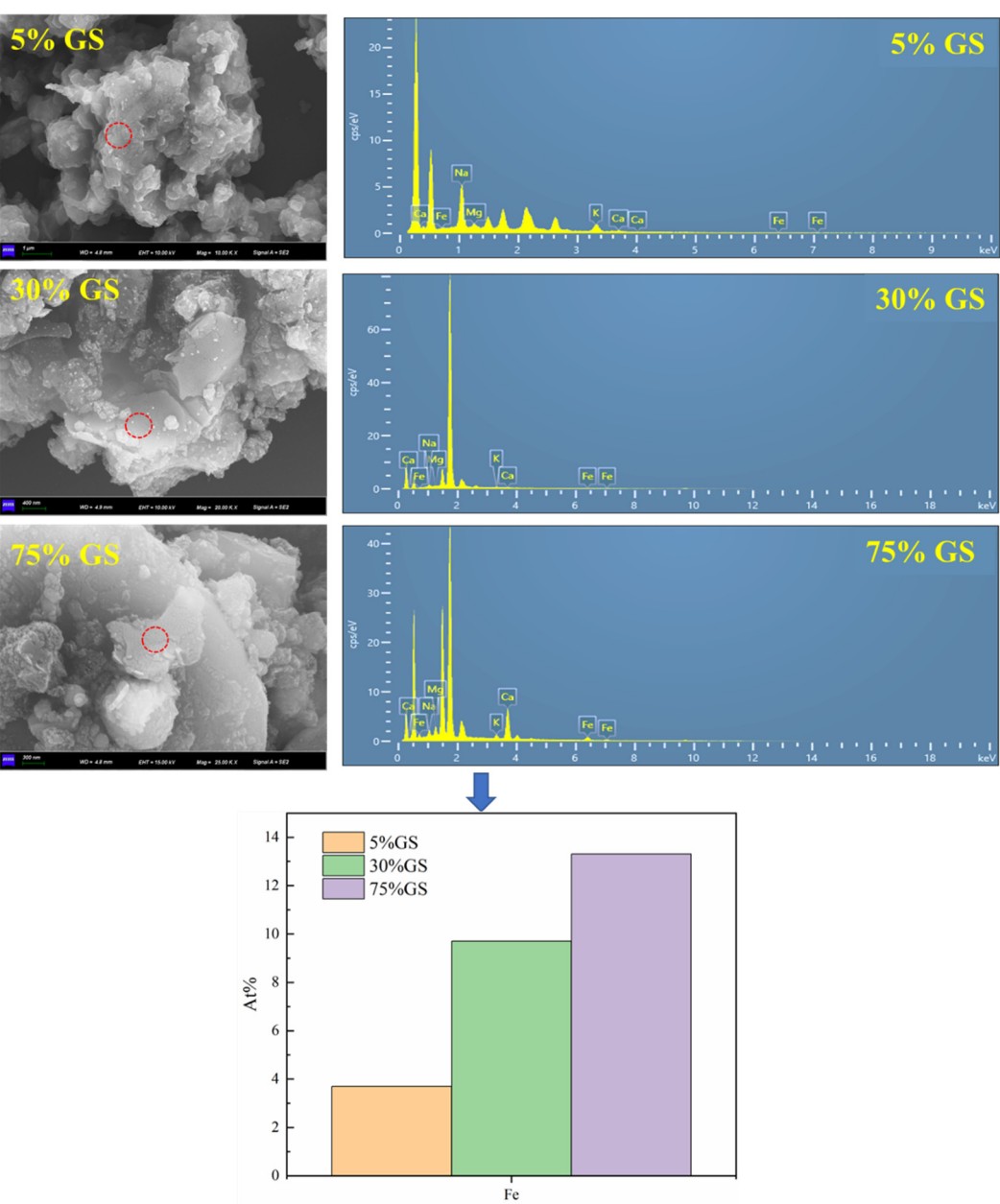

**Figure 5.** SEM-EDS images of 5% GS, 30% GS, and 75% GS cases.

## 3. Materials and Methods

### 3.1. Material Preparation

Coal gasification fine slag (GS), produced in the process of coal to liquids, was obtained from an entrained-flow gasification unit in State Energy Group Shenhua Ningxia Coal Industry Group Co. Ltd. Bituminous coal (BC) and bamboo residue (BR) were collected from Binzhou City, Shangdong Province and Yibin City, Sichuan Province in China, respectively. BRs are composed of bamboo branches and bamboo nodes generated in the process of cutting and utilizing. Before testing, GS and BC were dried at 105 °C in a vacuum-drying oven for 3 h. BR was dried at 85 °C for 3 h. The feedstock materials were pulverized and sieved to particle sizes less than 0.075 mm. The three samples were proportionally well blended in a ball mill at 400 r/min. The blending proportions are listed in Table 9.

**Table 9.** Experimental conditions.

| GS Mass Fraction/% | BR Mass Fraction/% | BC Mass Fraction/% |
| --- | --- | --- |
| 5 | 40 | 55 |
| 10 | 40 | 50 |
| 15 | 35 | 50 |
| 25 | 25 | 50 |
| 30 | 30 | 40 |
| 40 | 30 | 30 |
| 50 | 25 | 25 |
| 60 | 30 | 10 |
| 75 | 15 | 10 |

### 3.2. Analysis Methods

#### 3.2.1. Sample Properties

The basic properties of samples were detected by proximate analyses, ultimate analyses, and high heating values. The operation process was conducted based on the China National Standards GB/T 212-2008, GB/T 476-2008, and GB/T 213-2008, respectively.

#### 3.2.2. Ash Composition Analysis

The ash chemical compositions of samples were determined by XRF with the standard procedures of ASTM D4326. Prior to XRF analysis, the raw materials were ashed using a muffle furnace held for 1 h at 815 °C (BC and GS) and 550 °C (BR).

#### 3.2.3. Micromorphology Analysis

The microscopic morphology of the samples was tested through scanning electron microscopy (SEM, TESCAN VEGA COMPACT). Before the test, the sample was ultrasonically dispersed in anhydrous ethanol and then dropped on a slide. After the sample was completely dried, the sample powder was evenly adhered to the slide with conductive tape, and gold was sprayed to enhance the conductivity of the samples.

#### 3.2.4. Thermogravimetric Analysis

Thermogravimetry was widely used to study the thermochemical behaviors of solid fuels. The combustion of samples was evaluated using TGA (Netzsch STA 449 F5 Jupiter). Before testing, approximately 10 mg of the samples was placed in a corundum crucible. The samples were subjected to thermal decomposition at a heating rate of 10 °C/min from room temperature to 900 °C, and the total airflow rate was set to 20 mL/min.

### 4. Conclusions

In this paper, thermogravimetric analysis was applied to evaluate the co-combustion characteristics of BC, GS, and BR. It was found that the addition of BC and BR can improve the combustion performance of GS compared with individual combustion. The comprehensive combustion index S decreased gradually with the increase in GS added, suggesting that the combustibility of blends became worse. Notably, there were interactions between the tri-combustion process, both synergistic and antagonistic, via comparing the experimental and theoretical conversion. The synergistic effect was mainly caused by the catalysis of metals in the raw material. The interactions varied with different blending percentages of GS. The kinetic analysis indicated that the tri-combustion reaction was determined by the chemical reaction order and diffusion model (O1 and D3). The co-combustion active energy was lower than that of the individual combustion of GS. According to the activation energy of the first and second stages in the tri-blending combustion reaction and the synergistic effect, the blending ratio of 60% GS-30% BR-10% BC was recommended.

**Author Contributions:** Conceptualization, Y.Z., W.J. and J.W.; data curation, R.W.; funding acquisition, J.W.; investigation, Y.Z., W.J., R.W. and F.G.; methodology, B.D.; project administration, J.W.; software, Y.G.; visualization, Y.G.; writing—original draft, Y.Z. and W.J.; writing—review and editing, B.D. All authors have read and agreed to the published version of the manuscript.

**Funding:** This work was supported by the Fundamental Research Funds for the Central Universities (No. 2017QNA25).

**Data Availability Statement:** Not applicable.

**Conflicts of Interest:** The authors declare no conflict of interest.

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
