# Peer review of "Investigation of the Characteristics of Catalysis Synergy during Co-Combustion for Coal Gasification Fine Slag with Bituminous Coal and Bamboo Residue"

_catalysts, doi:10.3390/catal11101152_

Round 1

Reviewer 1 Report

I have read carefully the manuscript submitted by the authors. I agree with the methods applied, the discussion and, therefore, the conclusions obtained. 

I have only one question to the authors: what is the novelty of this work? No novelty on methods, no novelty on discussion, no novelty on results. All you have reported is well known between the broad literature about biomass and wastes combustion (or other valorization methods). 

It is not acceptable to publish results that don´t contribute to increase the technical or scientific knowledge of the field studied. In this case, the authors must work hard to justify where can I find a novel conclusion in this manuscript.

Author Response

We have tried our best to improve the manuscript and made changes to it. We also sincerely appreciate reviewers and editors for their time and consideration and hope that the revision will meet with approval. If there is anything else, we will be pleased to have the opportunity to revise it.

Once again, thank you very much for your kind work. I am looking forward to hearing from you.

Reviewer 2 Report

The topic of this manuscript is relevant to the scope of the Сatalysts. It deals with the study of the co-combustion behavior of the tri-fuel blends, including bituminous coal (BC), gasification slag (GS) and bamboo residue (BR), is observed with using  thermogravimetric analysis.. The TGA results show that the combustibility is increased owing to the addition of BC and BR, and the ignition and burnout tem-peratures are lower than those of GS alone. The combustion characteristics of the blended sample  become worse with an increase in the proportion of GS. The co-combustion process is divided into two main steps with obvious interactions (synergistic and antagonistic). The synergistic effect is mainly attributed to the catalysis of the ash-forming metals reserved with the three raw fuels. I think this manuscript may be published after minor revision as outlined below.

  1. It is desirable to explain how the presence of impurities (the metals in the raw materials) lead to decreasing of energy of activation.
  2. In what form is the metal in the samples of raw materials.
  3. Is it only an increasing of the amount of metal that leads to an increasing of  catalytic activity?

Author Response

(The authors gave the same response as above.)

Reviewer 3 Report

Comment and suggestions for authors in the file.

Author Response

(The authors gave the same response as above.)

Round 2

Reviewer 1 Report

The manuscript has been revised and the authors have highlighted some minor novelty in the actual form. Therefore, I recommend to accept it for publication in Catalysts.